# Lighting Enhancement and Skin Lesion Analysis in Macroscopic Images using Genetic Algorithms and Deep Neural Networks

Vanesa Gómez-Martínez[1] and Cristina Soguero-Ruiz[*][1]

[1]Department of Signal Theory and Communications, Rey Juan Carlos University, Madrid, 28943, Spain
{vanesa.gomez, cristina.soguero}@urjc.es

## 1 Introduction

Skin cancer is one of the most common cancers globally, with early detection crucial for better outcomes [1]. While dermoscopy is the gold standard [2], macroscopic imaging via smartphones offers a more accessible, low-cost alternative [3], though often affected by poor contrast, lighting, and sharpness—hindering automated diagnosis.

Existing methods like Intensity Equalization and Contrast Enhancement Techniques (IECET) [3] rely on paired images and focus mainly on brightness. We propose an unsupervised enhancement pipeline using genetic algorithms (GAs) [4] to optimize brightness, contrast, and sharpness without reference data. Enhanced images are processed via U-Net variants [5] for segmentation and ResNet-50 [6] for melanoma classification. We also apply Grad-CAM [7] for interpretability and bias analysis.

## 2 Materials and Methods

### 2.1 Dataset

We used a public dataset from the University of Waterloo [8, 9], containing 206 macroscopic skin lesion images (119 melanomas, 87 not melanomas) with expert-annotated segmentation masks.

### 2.2 Genetic algorithms for macroscopic image enhancement

GAs are unsupervised optimization methods inspired by natural selection [4]. In our approach, each chromosome encodes brightness, contrast, and sharpness parameters. A population of 50 chromosomes evolves over 40 generations, guided by our proposed fitness function $Q$, which combines three image quality metrics reflecting key aspects of medical image quality—enhancing lesion texture, borders, and tonal variation [10].

- **Contrast** ($C$): measures intensity variation between pixels and their neighbors, defined as: $C = \frac{1}{N_{\text{levels}}} \sum_{i=1}^{N_{\text{levels}}} \sum_{j=1}^{N_{\text{levels}}} (i-j)^2 \cdot P(i,j)$, where $P(i,j)$ is the gray-level co-occurrence matrix.

- **Brightness** ($B$): average intensity of the pixels in the image: $B = \frac{1}{N_{\text{pixels}}} \sum_{i=1}^{M} \sum_{j=1}^{N} I(i,j)$, where $I(i,j)$ is the grayscale value of pixel $(i,j)$.
- **Sharpness** ($Ni$): computed as the variance of the Laplacian: $Ni = \text{Var}(\text{Laplacian}(I))$.

The final quality score is computed as a normalized weighted sum: $Q = \frac{C_{\text{enh}}}{C_{\text{orig}}} + \frac{B_{\text{enh}}}{B_{\text{orig}}} + \frac{Ni_{\text{enh}}}{Ni_{\text{orig}}}$. Chromosomes with higher $Q$ values are favored during evolution, leading to optimal enhancement parameters.

### 2.3 U-Net and CNNs for skin lesion segmentation and classification

For skin lesion segmentation, we used the U-Net architecture and two variants: Attention U-Net (Att-Net) and Dense U-Net (D-Net) [5], which leverage an encoder-decoder structure to capture spatial and contextual features. Models were trained with Dice loss [11]. For melanoma classification, we employed ResNet-50 [6], a robust CNN for image tasks, trained with binary cross-entropy loss [12].

## 3 Results
### 3.1 Experimental setup

We conducted 5-fold cross-validation on the Waterloo dataset, using 80% of the data for training (15% for validation) and 20% for testing. Details of the experimental settings are summarized in Table 1.

**Table 1.** Summary of experimental setup.

| Component | Configuration |
| --- | --- |
| **Training parameters** | |
| GA | $P$=50, $G$=40, $C \in$[0.8, 1.2], $B \in$[−10, 10], $M$=20% |
| U-Net | Opt: Adam, BS: 16, ES: 55 epochs, loss: Dice |
| ResNet-50 | loss: BCE, ES: 30 epochs |
| **Evaluation metrics** | |
| Segmentation | Dice, IoU, Precision, AUCROC |
| Classification | AUCROC |

$P$ = population size, $G$ = generations, $C$ = contrast, $B$ = brightness, $M$ = mutation rate, Opt = optimizer, BS = batch size, ES = early stopping.

### 3.2 Illumination enhancement results

Figure 1 shows that IECET mainly improves brightness, but fails to enhance contrast or sharpness. In contrast, our GA-based method produces more balanced enhancements across all quality metrics. This is also confirmed in the 3D scatter plot (Figure 2).

*Corresponding Author.

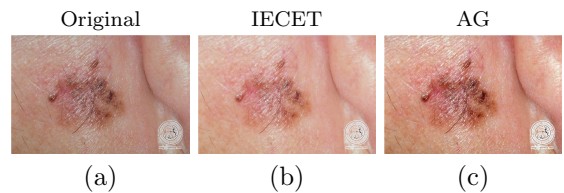

|     Original     |     IECET     |     AG     |
| :---: | :---: | :---: |
| (a) | (b) | (c) |

**Figure 1.** Visual example from the Waterloo dataset showing the effect of enhancement methods: (a) original, (b) IECET-enhanced, (c) GA-enhanced.

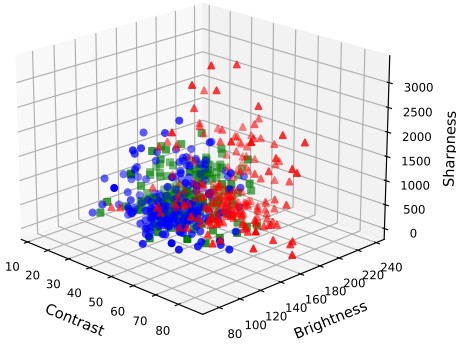

**Figure 2.** 3D scatter plot of image quality metrics (contrast, brightness, and sharpness) for three image sets: original (in blue), GA-enhanced (in red), and IECET-enhanced (in green).

## 3.3 Segmentation and classification performance

Table 2 summarizes segmentation metrics. GA-enhanced images combined with Attention U-Net yielded the best results, achieving a Dice score of 0.871 and AUCROC of 0.951. For melanoma detection, the best AUC-ROC (0.80) was achieved when using GA-enhanced and segmented images, as shown in Figure 3. This highlights the effectiveness of combining image enhancement and segmentation to improve classification performance.

**Table 2.** Segmentation metrics for U-Net variants using original, IECET-enhanced, and GA-enhanced images.

| Image type | Model | DI | IoU | Precision | AUCROC |
| :--- | :--- | :--- | :--- | :--- | :--- |
| Original | U-Net | 0.840±0.041 | 0.726±0.061 | 0.860±0.061 | 0.927±0.026 |
| | Att-Net | 0.868±0.035 | 0.769±0.054 | 0.896±0.043 | 0.944±0.022 |
| | D-Net | 0.866±0.032 | 0.765±0.050 | 0.881±0.040 | 0.937±0.019 |
| IECET | U-Net | 0.847±0.040 | 0.737±0.058 | 0.894±0.049 | 0.940±0.022 |
| | Att-Net | 0.851±0.040 | 0.745±0.065 | 0.899±0.039 | 0.943±0.021 |
| | D-Net | 0.846±0.036 | 0.735±0.064 | 0.875±0.052 | 0.932±0.025 |
| GA | U-Net | 0.859±0.035 | 0.755±0.053 | 0.881±0.021 | 0.935±0.011 |
| | Att-Net | **0.871±0.049** | **0.775±0.074** | **0.913±0.020** | **0.951±0.010** |
| | D-Net | 0.852±0.040 | 0.744±0.061 | 0.878±0.031 | 0.934±0.017 |

## 3.4 Interpretability analysis

To enhance transparency and interpretability, we used Grad-CAM to visualize CNN attention. As shown in Figure 4, original images (d) yield diffuse

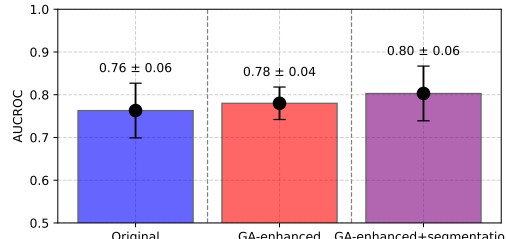

**Figure 3.** Mean±SD of AUCROC using ResNet-50 on original, GA-enhanced, and GA+segmentation images across 5 folds of the Waterloo dataset.

focus, often on irrelevant areas. GA-enhanced inputs (e) improve focus on lesions, and combining enhancement with segmentation (f) further sharpens attention, boosting relevance and interpretability.

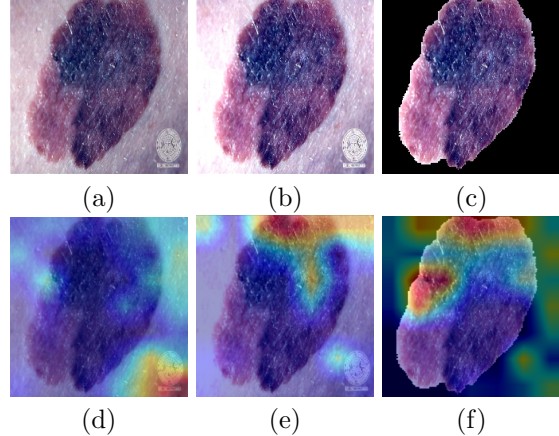

|     (a)     |     (b)     |     (c)     |
| :---: | :---: | :---: |
| (d) | (e) | (f) |

**Figure 4.** Example from the Waterloo dataset showing input types and their Grad-CAM heatmaps. Top row: (a) original, (b) GA-enhanced, (c) GA-enhanced + segmentation. Bottom row: corresponding Grad-CAM heatmaps for (d)–(f).

## 4 Conclusions

We propose a hybrid framework that combines unsupervised GA-based enhancement, U-Net segmentation, and ResNet-50 classification for macroscopic skin lesion analysis. Our method enhances image quality without reference data, tackling issues like low contrast and uneven lighting.

Experiments show improved segmentation across U-Net variants, with the best results from enhancement plus Attention U-Net. Melanoma detection also improves, achieving the highest AUCROC with GA-enhanced and segmented images. Grad-CAM analysis confirms better model focus on lesions, boosting interpretability and reducing irrelevant attention.

These results highlight the potential of our approach to support dermatological diagnosis using accessible imaging.

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
