# OpenReview forum: "Lighting Enhancement and Skin Lesion Analysis in Macroscopic Images using Genetic Algorithms and Deep Neural Networks"
_NLDL.org/2026/Abstracts_Track — NLDL 2026 Abstracts_

### Official Review · Reviewer_L87F · 2025-10-27

**Soundness:** 3
**Correctness:** 2
**Rating:** 4
**Confidence:** 4

**Summary:**

Authors propose an unsupervised enhancement pipeline using genetic algorithm to optimize at the same time brightness, contrast, and sharpness. In their experiments on the Waterloo dataset, GA enhanced images together with Att-Net yields the best results. Apparently, their strategy guides GRAD-CAM to focus on the right regions.

**Strengths:**

Overall, the abstract is well written and motivated, and it addresses a relevant topic with promising results shown in Table 2. The proposed method works without reference data. The main result is presented using four metrics, and the figures help convey the overall message of the paper. Although further studies are needed (e.g., additional datasets and more in-depth analyses; see the weaknesses section), I believe the abstract lays the foundations for an interesting and relevant paper

**Weaknesses:**

I am aware of the two-page limit and the intention to share new ideas with the community, but I would like to highlight a few points that deserve attention:
- W1: Figure 2. Although the idea can be understood, the density of the points makes it difficult to analyze performance. In particular, it may be clear that the red points stand out, but it is not clear whether they obtain a better overall score than their two counterparts (blue and green). It would be useful to include a numerical metric to confirm this. To be more precise: taken a red dot, if I don't know its counterparts (blue and green), how can I asses its performance? It can be the case of having 2 outstanding dimensions and 1 with poor performance, while the original part was more balanced (and the plot would not capture this aspect).
- W2: 3.4 and Conclusion. In lines 103-105 the authors state "Grad-CAM analysis confirms better model focus on lesions, boosting interpretability and reducing irrelevant attention.". While this may be true from Figure 4, no general conclusion can be drawn as the analysis is performed on a single instance.
- W3: In figure 4 it is implicitly assumed that Grad-CAM is faithful (i.e., it reflects the decision process of the model), which is not always the case.
- W4: Figure 3. It takes up a lot of space with a very short message. Numerical values or a more complete table, such as Table 2, would have been better choices.

Moreover, as questions:
- Q1: How would you explain that GA with D-Net is showing a worse performance compared to the original?
- Q2: With ResNET-50 there is no significant improvement on AUCROC, what about the other three metrics?
- Q3: What do DI and IoU stands for in Table 2?

---

### Official Review · Reviewer_V89f · 2025-11-02

**Soundness:** 3
**Correctness:** 3
**Rating:** 2
**Confidence:** 4

**Summary:**

The paper presents a genetic algorithm for unsupervised enhancement of images of skin lesions. The algorithm is intended as a preprocessing step to optimize image brightness, contrast and sharpness prior to segmentation and/or melanoma classification. The approach is especially intended for smartphone images where quality may be low. Experiments are carried out with and without preprocessing in combination with different segmentation methods (Unet, AttNet, Dnet) and for melanoma classification with a ResNet50 model, comparing performance. The results in general show some improvement when using the preprocessing.

**Strengths:**

The abstract is clear and easy to follow with good illustrations and tables summarizing results.

**Weaknesses:**

The idea of using genetic algorithms for image enhancement is not very new, although it may not have been applied in this exact context. Furthermore, for a deep learning conference it would have been good to see more focus on the deep learning algorithms, especially as the choice of segmentation algorithm seems to have a much greater impact on performance than the effect of preprocessing.
For figure 3 it would have been good to see the classification performance also for the segmentation of the original, and it is not totally clear from the text which segmentation algorithm that was used (assume attention Unet).
As the focus is on image enhancement, it would have been good to see some more information on the image quality of the dataset that was used and whether these originate from smartphones.

---

### Official Review · Reviewer_8Mjg · 2025-11-03

**Soundness:** 3
**Correctness:** 2
**Rating:** 4
**Confidence:** 3

**Summary:**

The abstract presents a hybrid framework for deep-learning-based skin lesion segmentation and melanoma classification. It features an image enhancement mechanism based on a genetic algorithm that does not require expert labels. The framework enhances the contrast and lighting of the processed images, and achieves a dice score of 0.871 for segmentation, and an AUCROC of 0.8 for classification on the Waterloo dataset.

**Strengths:**

1. __Writing and presentation.__ The abstract is well written and easy to follow.
2. __GA-based enhancement.__ The authors propose a genetic algorithm for image enhancement, which does not require explicit expert labels.
3. __Complete method description.__ The authors explain the fitness function, network- and training details and used data.
4. __Interpretability analysis.__ The authors provide a qualitative interpretability analysis, which is crucial for discussion with biomedical professionals.

**Weaknesses:**

1. __Limited dataset and weak generalization.__ The Waterloo dataset (206 images) is far too small for training and evaluating deep models like ResNet-50 or U-Net. The reported improvements (Dice = 0.871, AUC = 0.80) may not generalize beyond this small sample.
→ Recommendation: Validate on larger, public datasets (e.g., ISIC, HAM10000) to demonstrate robustness.
2. __Questionable “unsupervised” claim.__ The authors refer to their enhancement process as unsupervised, but the optimization is driven by a manually defined quality metric (brightness, contrast, sharpness). This is more accurately described as a heuristically guided optimization, not true unsupervised learning.
→ Recommendation: Consider task-driven optimization (e.g., using segmentation accuracy as a fitness signal).
3. __Insufficient comparison with modern enhancement techniques.__ The only baseline used is IECET [3], a basic intensity equalization method. Recent deep-learning-based approaches (e.g., RetinexNet, EnlightenGAN, Zero-DCE, diffusion-based enhancement) should be included to strengthen the evaluation.
→ Recommendation: Add comparisons against at least one deep-learning-based low-light enhancement model.
4. __Interpretability analysis is superficial.__
Grad-CAM visualizations are shown qualitatively, but no quantitative assessment of attention alignment is provided. This makes the interpretability claim weak.
→ Recommendation: Quantify lesion-focused attention overlap between Grad-CAM maps and segmentation masks.
5. __No analysis of computational efficiency.__ Genetic algorithms are computationally expensive. Optimizing per image across 40 generations × 50 chromosomes could be impractical for real-time or mobile deployment.
→ Recommendation: Include runtime analysis and discuss scalability or propose a neural surrogate model for deployment.
6. __Overstated clinical relevance.__
The abstract suggests potential use in dermatological diagnosis, but no clinician-based validation or user study is performed.
→ Recommendation: Temper claims or include preliminary expert feedback on clinical usefulness.

---

### Decision · Program_Chairs · 2025-11-05

**Decision:**

Accept

**Comment:**

The reviewers found the abstract borderline, yet the PCs believe it will be of interest to the community and should have the opportunity be presented.